# Chemical Dissolution-Assisted Ultrafine Grinding for Preparation of Quasi-Spherical Colloids of Zinc Oxide

**DOI:** 10.3390/ma16072558

**Published:** 2023-03-23

**Authors:** Guanghua Huang, Zening Chen, Zhidong Pan, Yan Xu, Hanlin Hu, Yanmin Wang

**Affiliations:** 1Postdoctoral Innovation Practice Base of Hoffman Institute of Advanced Materials, Shenzhen Polytechnic, Shenzhen 518055, China; 2School of Materials Science and Engineering, South China University of Technology, Guangzhou 510640, China

**Keywords:** zinc oxide, stirred media mill, quasi-spherical particles, chemical dissolution, grinding mechanism

## Abstract

Submicron-sized quasi-spherical zinc oxide (ZnO) particles were prepared by wet ultrafine grinding in a stirred media mill under various conditions. The effects of parameters (i.e., solution type, acid or alkali concentration, solid content and grinding time) on the particle median size (d_50_), particle size distribution (PSD) and sphericity of ZnO particles was investigated. The results show that submicron-sized quasi-spherical particles (i.e., d_50_: 370 nm, uniformity coefficient (*n*) of 2.28 and sphericity of 0.91) can be obtained when the micron-sized ZnO particles are ground for 30 min in a CH_3_COOH solution at a concentration of 0.010 mol/L with 20 wt.% of solid content. The chemical dissolution of ZnO particles ground in the presence and absence of acetic acid is discussed. It is indicated that chemical dissolution accelerated due to the mechanochemical effects could reduce the particle size, obtain a narrower PSD and enhance the sphericity. In addition, the functions of selection and breakage were used to analyze the grinding mechanism of ZnO particles.

## 1. Introduction

Submicron-and nano-sized zinc oxide (ZnO) particles are widely used in diverse applications, such as electronics, biomedicine, coating, fillers, catalysts and cosmetics, due to their unique electrical, optical and thermal properties [1,2,3,4,5,6]. In addition to their fineness, the morphology (i.e., sphericity) of ZnO particles has an important effect on its performances in subsequent applications [4]. Haile et al. [7] reported that spherical ZnO particles are suitable for varistors electronics due to their high density and unique electrical properties. Yuan et al. [8] also reported that an electronic ceramic with the homogeneous microstructure and improved performance can be prepared with fine spherical ZnO particles.

The existing methods available for the preparation of submicron-and nano-sized spherical or quasi-spherical ZnO particles are chemical and physical approaches. The chemical methods include precipitation [9], sol–gel [10], hydrothermal [11], emulsion [12], mechanochemical process [13] and spray pyrolysis [14] techniques. The physical methods are mainly mechanical ultrafine grinding, plasma heating and laser ablation techniques [3,15,16]. Among the methods above, the spherical properties prepared by plasma heating and laser ablation have the advantages of high sphericity and uniform size distribution. However, the equipment and production costs required by these two methods are large and the output is small, so they cannot be used in industry and mass production. Mechanical ultrafine grinding has the advantages of low costs, a simple process and being suitable for mass production. It is simply a method that applies a certain sort of mill-like stirred media mill. As everyone knows, the grinding mechanism in a stirred media mill is mainly due to the shear and compressive stresses between beads as grinding media, which have an abrasive effect on the particles’ surfaces. As a result, irregularly shaped particles ground in a stirred media mill become more regular, eventually having less sharp corners and edges [17]. In addition, ZnO is a kind of amphoteric oxide that can react with acid or alkali. The chemical dissolution on a ZnO particle’s surface is affected by pH value, acid or alkali concentration, ionic strength, surface morphology, crystal structure and temperature [18,19,20]. The dissolution reaction could be due to the direct proton attacks on the particle’s surface or the formation of complexes [21,22]. It is thus presumed that quasi-spherical, ultrafine ZnO particles could be prepared by wet ultrafine grinding in a stirred media mill with a proper assistance from chemical dissolution. Chen et al. [23] prepared silica particles ground within a low-concentration solution of barium chloride in a stirred media mill, and found that the chemical dissolution in a short period of grinding could improve the particles’ morphology and narrow the PSD. In the presence of barium chloride, they also consider that the chemical dissolution of silica on the particles’ surfaces could be accelerated due to the mechanochemical effect of the mill.

Hence, submicron-sized quasi-spherical ZnO particles were prepared in a stirred media mill within various solutions (i.e., water, oxalic acid dihydrate ((COOH)_2_·2H_2_O), acetic acid (CH_3_COOH), hydrochloric acid (HCl), citric acid (C_6_H_8_O_7_), ammonium, mixed hydroxide/ammonium chloride and sodium hydroxide). The effects of solution type, acid or alkali concentration, solid content and grinding time on the d_50_, PSD and sphericity of ZnO particles were investigated. The chemical dissolution of particles ground in the mill with and without chemical-dissolution assistance was discussed. Furthermore, the grinding mechanism of ZnO particles in the absence and presence of acetic acid was also analyzed by the functions of selection and breakage.

## 2. Experimental

### 2.1. Materials

Micron-sized ZnO with fineness of 95 wt.% < 5 μm and sphericity of 0.73 was used as a raw material. It was produced by a pyrometallurgical method (Huizhou 74 Huanmeisheng Novel Material Ltd., China). Chemical reagents used were acetic acid (CH_3_COOH, ≥99.5%), hydrochloric acid (HCl, 37%), oxalic acid dihydrate ((COOH)_2_·2H_2_O, ≥99.5%), citric acid (C_6_H_8_O_7_, ≥99.5%), ammonium hydroxide (NH_4_OH, 25–28%), sodium hydroxide (NaOH, 96%) and ammonium chloride (NH_4_Cl, ≥99.5%). All materials used in this paper was not further purified.

### 2.2. Preparation

Wet ultrafine grinding of ZnO particles in water and different solutions was performed in a model W-0.1 vertical stirred bead mill (Shenzhen Sanxing Feirong Machine Ltd., Shenzhen, China) with zirconia beads of 0.6–0.8 mm in diameter at a fixed rotational speed of 1500 rpm for different periods of time (i.e., 30, 45 and 60 min). The stirred bead mill was operated in a continuous mode and cooled to keep a constant temperature. Table 1 shows the different experimental factors and levels during the experiment. The solutions of HCl, CH_3_COOH, (COOH)_2_·2H_2_O, C_6_H_8_O_7_, NH_4_OH/NH_4_Cl and NaOH at different low concentrations (i.e., 0.005, 0.010, 0.050 and 0.100 mol/L, respectively) were used in wet ultrafine grinding. The solid contents of ZnO particles were 20, 30 and 40 wt.%. In each experiment, the total mass of zinc oxide suspension was 125 g.

### 2.3. Characterizations

The PSD of ZnO particles were analyzed by a model BT-9300S laser diffraction particle size analyzer (Dandong Bettersize Instruments Co., Dandong, China). The morphology of particles was determined by a model NOVA NANOSEM 430 scanning electron microscope (SEM) (Thermo Fisher Scientific Co., Waltham, MA, USA). The specific surface area was tested by a model Flowsorb3 2310 nitrogen adsorption BET device (Micromeritics Instruments Co., Norcross, GA, USA). A model Optima 8300 inductively coupled plasma optical emission spectroscope (ICP-OES) (Perkin Elmer Instruments Co., Waltham, MA, USA) was applied to detect the concentration of zinc ions in the aqueous solution. A model PB-10 pH-meter (Sartorius Co., Hamburg, Germany) was used to measure the pH value of the solution before and after grinding/stirring. The changes in crystallinity of samples being ground were tested by a PW-1710 X-ray diffractometer (PANalytical Co., Eindhoven, The Netherlands).

The sphericity of ZnO particles was analyzed on the SEM images by Image-Pro Plus 6.0 image analysis software (Media Cybernetics, Inc., Rockville, MD, USA). The sphericity (*Φ*) of each particle was calculated by
(1)$$\Phi =\frac{4\pi S}{p^{2}}$$
where *p* is the particle projection perimeter and *S* is the particle projection area. The maximum value of *Φ* is 1; the greater the *Φ* value is, the more spherical the particles will be. In this paper, the average sphericity was determined by calculating the sphericities of 100 particles according to Equation (1).

The scale of the PSD was tested by the Rosin–Rammler–Bennett (RRB) model [24]:(2)$$R_{\mathrm{RRB}}=100\exp[-(\frac{d}{d_{e}}{)}^{n}]$$
where *R_RRB_* is the volume percentage of the particles with diameter > *d*, *d* is the particle diameter, *d_e_* is the size modulus (i.e., the diameter when *R_RRB_* = 36.8%) and *n* is the uniformity coefficient. The greater the value of *n* is, the narrower the PSD curve will be Equation (2) can be expressed as
(3)$$\ln(\ln\frac{100}{R_{\mathrm{RRB}}})=n(\ln d-\ln d_{e})$$

The data of PSD at 5–95% were selected for lined fitting so as to reduce the error of linear fitting.

### 2.4. Simulation by Modeling

In this paper, the breakage behavior of zinc oxide particles was analyzed by the population balance model (PBM) in a stirred media mill. According to Reid [25], a practical approximation of the fundamental integro-differential PBM equation for batch grinding is proposed:(4)$$\frac{dN_{i}(t)}{dt}=-S_{i}N_{i}(t)+\sum_{j=i}^{\infty }S_{i}b_{i,j}N_{j}(t)$$
where *i* is the size class, *t* is the grinding time, *N_i_*(*t*) is the mass fraction of particles in the size class at grinding time *t*, *S_i_* is the selection function, *b_i,j_* is the breakage function and *I* is the size class of particles fractured from size class *j* (*i* ≥ *j*). The range of particle size can be divided into *n** classes in geometric progression from 1 (coarse) to *i* (fine), and it is assumed that the grinding probability of particles in the same size class is equal. The selection function (*S_i_*) represents the breakage probability per unit time of particles in size class *i*. It reflects the difficulty of particles breaking in each size class. The breakage function (*b_i,j_*) represents the size distribution of daughter particles from mother particles; it describes the form of the resulting particle size distribution.

The transformation in a cumulative fraction residual mode of Equation (4) is [26]
(5)$$\frac{dR_{i}(t)}{dt}=-S_{i}R_{i}(t)+\sum_{j=1}^{i-1}[\left(S_{j+1}b_{i,j+1}-S_{j}b_{i,j})R_{j}(t\right)]$$
where *R_i_*(*t*) is the cumulative mass fraction of size class *i* at grinding time *t*. The solution of Equation (6) can be used to predict the size distribution in the batch grinding process. The approximate solution is
(6)$$\ln\frac{R_{i}(t)}{R_{i}(0)}\approx {K_{i}}^{\left(1\right)}t$$
where *K_i_*^(1)^ is the first-order Kapur function of size class *i*, which can be calculated via linear fitting. The calculation equations of *S_i_* and *b_i,j_* are
(7)$$S_{i}=-{K_{i}}^{\left(1\right)}$$
(8)$$b_{i,j}=\frac{K_{i-1}^{\left(1\right)}-{K_{i}}^{\left(1\right)}}{{K_{j}}^{\left(1\right)}}$$

## 3. Results and Discussion

### 3.1. Wet Ultrafine Grinding without and with Chemical Dissolution

Figure 1 shows the PSD of ZnO particles ground in water and different chemical solutions (i.e., HCl, CH_3_COOH, (COOH)_2_·2H_2_O, C_6_H_8_O_7_, NH_4_OH/NH_4_Cl and NaOH) at a rather low concentration of 0.010 mol/L for 30 min. Figure 1a and b show the interval mass distribution and cumulative mass distribution of ZnO particles, respectively. Table 2 shows the uniformity distribution coefficient (n), sphericity and corresponding median size (d_50_) of ZnO particles ground for 30 min in different solutions. In Figure 1, compared to the feed particles (i.e., d_50_: 639 nm, n: 1.38 and sphericity: 0.73), ZnO particles ground in water become finer and the PSD became more or less narrow. The d_50_ value decreased to 453 nm, the n value increased to 1.69 and the sphericity increased slightly to 0.78. This indicates that the grinding in a stirred media mill can reduce the particle size, obtain a narrow PSD and increase sphericity without chemical dissolution. The sphericity enhancement could be because the particles are subject to abrasion and tend to become more or less rounded in shape [27]. It is also interesting that the PSD of ZnO particles ground in chemical solutions is narrower than that of ZnO particles ground in water, and the particles are finer, indicating that the chemical solutions used in grinding could further improve the product’s quality (i.e., particle size/size distribution and sphericity). In Table 2, unlike having water as a solution in grinding, chemical solutions used in ultrafine grinding as ‘grinding aids’ all have a positive effect on the product’s quality, i.e., decreasing the d_50_, increasing the n and increasing the sphericity to different degrees. For the particles ground in the CH_3_COOH solution, the d_50_ value decreased to 370 nm, the n value increased to 2.28 and the sphericity increased to 0.91. These results suggest that submicron-sized quasi-spherical ZnO particles could be obtained when grinding in a solution with organic acid (i.e., CH_3_COOH, (COOH)_2_·2H_2_O or C_6_H_8_O_7_) rather than with inorganic acid (i.e., HCl) or base (i.e., NaOH) at a low concentration.

The coarse ZnO particles reduced in size distinctly after being ground in the mill for 30 min. Coarse ZnO particles have plenty of internal cracks, a more unstable structure and low surface energy; thus, the breakage needs lower energy and takes place readily. However, it is more difficult to fracture finer ZnO particles in the mill. According to Gao et al. [28], in a stirred media mill, the particles are compressed due to the centrifugal effect and generate some internal cracks, so cleavage occurs and the initial particles become medium-sized particles. In addition, shear stress between beads as grinding media, acting on the particles’ surfaces, cause particle abrasion; consequently, the size of finer ZnO particles decreases slightly during the grinding process. The ZnO particles ground for 30 min in water were comprised of mainly coarse particles and fine particles, along with a small portion of medium-sized particles, showing that shear and compressive stresses are the dominant forces for the particle breakage in a stirred media mill. Note that the average size of particles ground in a chemical solution (especially in CH_3_COOH solution) is lower (compared to water grinding), and the portion of finer particles increases significantly, indicating an effect of chemical dissolution on the particles’ surfaces in stirred media grinding.

Figure 2 shows the SEM pictures of feed and ZnO particles ground for 30 min in water or CH_3_COOH solution. In Figure 2a, the feed particles have a wider size range with a great portion of coarse particles and irregular morphologies with sharp corners and edges. In Figure 2b, the ZnO particles ground for 30 min in water become finer with improved sphericity (see Table 2), indicating that ultrafine grinding of ZnO in a stirred media mill could reduce the particle size and slightly increase the sphericity as well. In Figure 2c, the ZnO particles ground in a CH_3_COOH solution at a low concentration of 0.010 mol/L have a finer particle size and a quasi-spherical morphology without obvious corners and edges. This indicates that the use of CH_3_COOH in ultrafine grinding could result in smaller particles, a narrower PSD and enhanced sphericity. That is, the chemical dissolution on the particles’ surfaces during ultrafine grinding in stirred media mill has a positive effect on particle quality (i.e., d_50_, PSD and sphericity).

Table 2 shows the d_50_, *n*, sphericity and specific surface area of ZnO particles ground for 30 min in water or one of various solutions at a concentration of 0.010 mol/L. In Table 2, the feed particles exhibit a low specific surface area of 3.55 m^2^/g, and the specific surface area of particles ground for 30 min in water or in CH_3_COOH solution increased to 9.77 or 11.39 m^2^/g. The increment in specific surface area also shows the particle-size reduction and sphericity enhancement.

Figure 3 shows the PSD and SEM pictures of ZnO particles stirred or ground for 30 min in a CH_3_COOH solution at a low concentration of 0.010 mol/L. In Figure 3, compared to the particles in feed (i.e., d_50_: 639 nm), the d_50_ of the particles stirred in the CH_3_COOH solution slightly decreased to 603 nm, and the particles still maintained irregular shapes with obvious corners and edges. This indicates that chemical dissolution in the CH_3_COOH solution without mechanical grinding was inefficient. The particles ground in a CH_3_COOH solution at a low concentration of 0.010 mol/L become finer and more uniform. They had a quasi-spherical morphology. This phenomenon could be possibly attributed to the mechanochemical effects of the stirred media mill. Romeis et al. [29] reported that the mechanochemical effects could increase the reactivity of particles in a stirred media mill, and the liquid phase influences the particles’ size and shape. The chemical dissolution on particle surfaces was more effective due to the mechanochemical effects, leading to the particle size reduction and the increase in sphericity. Since some corners and edges of the particles have greater defect densities than smooth places on the particles’ surfaces, the chemical dissolution reaction in an acidic solution occurs even more dramatically in such places [30]. The corners and edges of particles are more easily rubbed round by shear stress between beads in the mill. Thus, wet ultrafine grinding in an acetic acid solution can fabricate submicron-sized ZnO particles with a narrow PSD and a quasi-spherical morphology.

Figure 4 shows the PSD of ZnO particles ground for 30 min in CH_3_COOH, C_6_H_8_O_7_, HCl and NaOH at different concentrations. The corresponding d_50_, *n* and sphericity are shown in Table 3. Clearly, compared to the particles ground in C_6_H_8_O_7_, HCl or NaOH solutions, the particles ground in the CH_3_COOH solution at an optimum concentration of 0.010 mol/L had a finer particle size (i.e., d_50_ = 370 nm), a narrower size distribution (n = 2.28) and a higher sphericity value (i.e., 0.91). The particles ground in other solutions at the concentration of 0.100 mol/L were larger in size, had a wider size distribution and had less sphericity. In these cases, the coarser particles remained and even increased in number at a higher concentration. This phenomenon could be caused by the particle aggregation caused at a lower pH value and a higher ion concentration [18]. The particle aggregation could affect the grinding performance and weaken the effect of chemical dissolution. A lower concentration of acid can promote the grinding and the dissolution efficiency. The particles ground in the C_6_H_8_O_7_ solutions at different concentrations also had a finer size and a narrower PSD, and the sphericity increased slightly as the concentration of C_6_H_8_O_7_ increased. This is because citrate anions adsorbed on the particle’s surface can increase the suspension stability [31] and also the dissolution effect of C_6_H_8_O_7_, thereby enhancing the grinding performance. In Figure 4d, the NaOH concentration is shown to have a slight influence on the grinding performance.

Figure 5 shows the PSD of ZnO particles ground for different lengths of time in water or in a CH_3_COOH solution at a low concentration of 0.010 mol/L. The corresponding d_50_, *n* and sphericity are show in Table 4. Clearly, the particles ground in water became finer and the sphericity increased as the grinding time went from 30 to 60 min. This is because particles are subjected to shear stress, and the edges and corners of particles ground for a longer period of time are abraded more. The sphericity of particles ground in the CH_3_COOH solution decreased as the grinding time was extended from 30 to 60 min. This could have been because the CH_3_COOH reacted completely after 30 min and the chemical dissolution ceased. The sphericity of particles is mainly influenced by mechanical grinding. In addition, the size/size distribution of the particles ground in the CH_3_COOH solution for 30 min was even finer/narrower than that of the particles ground in water for 60 min. This shows that the addition of CH_3_COOH in the solution has auxiliary effect on particle size reduction, particle size distribution narrowing, and increasing particle sphericity during ultrafine grinding in stirred media mill.

Figure 6 shows the PSDs of ZnO particles ground for 30 min in a CH_3_COOH solution at a low concentration of 0.010 mol/L with different solid contents. The corresponding d_50_, *n* and sphericity values are shown in Table 5. Apparently, the particles ground in the CH_3_COOH solution with a solid content of 20 wt.% had a finer median size (d_50_: 370 nm), a narrower PSD and a higher sphericity value (0.91). In general, the particles are more likely to aggregate in a denser suspension, and the contact probability among the particles and beads will reduce as the solid content increases. Additionally, the aggregation of particles will consume part of the energy for dispersion, which will also reduce the grinding efficiency. In addition, the amount of CH_3_COOH in the solution is reduced relatively when there is a higher solid content, thereby weakening the chemical dissolution’s effect on the grinding performance.

### 3.2. Chemical Dissolution Due to Mechanochemical Effect

It is necessary to clarify the chemical dissolution kinetics of ZnO particles ground in a stirred bead mill for a short time with respect to the mechanochemical effect. The concentrations of Zn^2+^ ions (c(Zn^2+^)) and pH values of ZnO suspensions ground in different solutions for different lengths of time can reflect a hybrid effect of chemical dissolution and shear/compressive stresses of beads on the particle surfaces in grinding. The chemical dissolution could occur on the corners/edges and surfaces of particles, hence decreasing the particle size, providing a narrower PSD and increasing the sphericity. Zn^2+^ ions generated due to the chemical dissolution on the particles ground or stirred in a solution can reflect the dissolution degree at a given point in time. The suspension’s pH value is another indication.

Figure 7 shows the Zn^2+^ ion concentrations and pH values of ZnO particles ground for different lengths of time in solutions of CH_3_COOH at different concentrations. In Figure 7, the suspension ground in water has few Zn^2+^ ions and has higher pH values due to the hydrolysis of ZnO particles [32], and the particles ground in solutions with various concentrations of CH_3_COOH have higher Zn^2+^ ions concentrations and lower pH values. This indicates that the use of CH_3_COOH in grinding can promote the particles’ dissolution. For the particles ground in the 0.010 mol/L CH_3_COOH solution, the c(Zn^2+^) and pH value varied rapidly from 0 to 10 min and slightly changed from 20 to 30 min, implying the chemical dissolution occurs readily in the early stage of grinding, and the dissolution is almost finished after 30 min. For the particles stirred in a 0.010 mol/L CH_3_COOH solution, the c(Zn^2+^) and pH value varied mildly and tended to increase after 30 min of stirring. The dissolution variation in the particles ground and stirred in a 0.010 mol/L CH_3_COOH solution indicates that the grinding process can accelerate the particles’ dissolution significantly due to the possible mechanochemical effect.

The dissolution of ZnO particles is because protons attack the surface Zn-O bonds directly. The dissolution reaction of ZnO particles in acid follows [22].
(9)$$ZnO_{\left(S\right)}+2H^{+}\to {Zn}^{2+}+H_{2}O_{\left(l\right)}$$

A slow dissolution of ZnO particles’ surfaces without grinding could occur. During grinding, however, the particle breakage leads to a greater surface area, thereby increasing the possibility of protons’ attack on the surface Zn-O bonds and accelerating the chemical dissolution on the particles.

In addition, Figure 8 shows the XRD patterns of feed and zinc oxide particles ground for 30 min in water or in a CH_3_COOH solution at a low concentration of 0.010 mol/L. Obviously, for particles ground in water or in a CH_3_COOH solution, the intensities of diffraction peaks are lower, and the lines of diffraction peaks are broader, especially for the particles ground in the CH_3_COOH solution. This indicates that the breakage of particles during grinding increases the surface-defect density and creates more hot spots for the particle surfaces’ dissolution [30,33]. The breakage of particles and chemical dissolution on the particles’ surfaces lead to the size reduction and morphology change, and hence, the particles’ surface dissolution in grinding due to the mechanochemical effect could affect the particle size, PSD and sphericity.

### 3.3. Simulation and Grinding Mechanism

The grinding mechanism of ZnO particles can be analyzed by the selection (*S_i_*) and breakage (*b_i,j_*) functions in PBM (see Equations (7) and (8)). It is proposed that the total grinding time should be 30 min, and ZnO particles were sampled at 5 min intervals in the grinding process. The particle size range was divided into 10 size classes (see Table 6).

Figure 9 shows the first-order Kapur functions calculated by linear fitting of the cumulative-fraction residual of particles ground in water and in the CH_3_COOH solution. In Figure 9, the model’s fitting lines agree with the experimental data properly. Hence, the PBM could be used in the particle system. The absolute values of the slopes of fitting lines are the selection-function values of different size classes, which represent the amounts of ZnO particles ground per unit time. That is, if the absolute value of the slope increases, more particles in the corresponding size class will be broken, which is based on a larger value of the selection function. Clearly, for the particles ground in water or in a CH_3_COOH solution, the coarser particles have steeper lines (i.e., particle size > 850 nm) and the finer particles less-steep lines (i.e., particle size < 361 nm). This indicates that the difficulty of breakage increases as the particle size reduces; the finer particles are difficult break. This is because that coarser particles are easily fractured by extrusion of the grinding medium; however, finer particles have fewer internal cracks, a more stable structure and greater surface energy, consequently requiring greater energy input for fracturing [34,35]. In addition, the selection-function values of different size classes of zinc oxide particles ground in a CH_3_COOH solution are greater than those ground for the particles ground in water, especially for the coarser particles (i.e., particle sizes > 1054 nm). However, the solution’s effect (CH_3_COOH or water) reduces as the particle size decreases. The effectiveness of these solutions at grinding the finer particles (i.e., particle sizes < 235 nm) is similar. This indicates that ultrafine grinding in a CH_3_COOH solution could reduce the particle size and improve the size distribution, and the solution is especially effective effect at the size reduction of the coarser particles.

Equations (10) and (11) are the breakage-function matrices of particles ground in water or in the CH_3_COOH solution, respectively. In Equations (10) and (11), for the particles ground in water and in the CH_3_COOH solution, the portions of particles in coarser size classes and finer size classes are greater than those in the medium size classes. This relative size distribution of the particles conforms to the case of an abrasive function, indicating that the primary stress is shearing and the secondary one is compression [28,34]. In addition, the particles ground in the CH_3_COOH solution have larger portions of particles in low and high size classes, implying that the shear stress has a dominant effect on the particles ground in the CH_3_COOH solution rather than in water. Hence, the assistance of chemical dissolution in wet ultrafine grinding can reduce the particle size.


(10)
$$B_{\mathrm{water}}=\left(\begin{array}{cccccccccc} 0 & 0 & 0 & 0 & 0 & 0 & 0 & 0 & 0 & 0\\ 0.3139\; & 0 & 0 & 0 & 0 & 0 & 0 & 0 & 0 & 0\\ 0.2051\; & 0.2989\; & 0 & 0 & 0 & 0 & 0 & 0 & 0 & 0\\ 0.1418\; & 0.2066\; & 0.2947\; & 0 & 0 & 0 & 0 & 0 & 0 & 0\\ 0.1063\; & 0.1550\; & 0.2211\; & 0.3134\; & 0 & 0 & 0 & 0 & 0 & 0\\ 0.0835\; & 0.1218\; & 0.1737\; & 0.2463\; & 0.3587\; & 0 & 0 & 0 & 0 & 0\\ 0.0557\; & 0.0812\; & 0.1158\; & 0.1642\; & 0.2391\; & 0.3729\; & 0 & 0 & 0 & 0\\ 0.0481\; & 0.0701\; & 0.1000\; & 0.1418\; & 0.2065\; & 0.3220\; & 0.5135\; & 0 & 0 & 0\\ 0.0253\; & 0.0369\; & 0.0526\; & 0.0746\; & 0.1087\; & 0.1695\; & 0.2703\; & 0.5556\; & 0 & 0\\ 0.0203\; & 0.0295\; & 0.0421\; & 0.0597\; & 0.0870\; & 0.1356\; & 0.2162\; & 0.4444\; & 1 & 1 \end{array}\right)$$



(11)
$$B_{{\mathrm{CH}}_{3}\mathrm{COOH}}=\left(\begin{array}{cccccccccc} 0 & 0 & 0 & 0 & 0 & 0 & 0 & 0 & 0 & 0\\ 0.3595\; & 0 & 0 & 0 & 0 & 0 & 0 & 0 & 0 & 0\\ 0.2117\; & 0.3305\; & 0 & 0 & 0 & 0 & 0 & 0 & 0 & 0\\ 0.1387\; & 0.2165\; & 0.3234\; & 0 & 0 & 0 & 0 & 0 & 0 & 0\\ 0.0985\; & 0.1538\; & 0.2298\; & 0.3396\; & 0 & 0 & 0 & 0 & 0 & 0\\ 0.0675\; & 0.1054\; & 0.1574\; & 0.2327\; & 0.3524\; & 0 & 0 & 0 & 0 & 0\\ 0.0456\; & 0.0712\; & 0.1064\; & 0.1572\; & 0.2381\; & 0.3676\; & 0 & 0 & 0 & 0\\ 0.0401\; & 0.0627\; & 0.0936\; & 0.1384\; & 0.2095\; & 0.3235\; & 0.5116\; & 0 & 0 & 0\\ 0.0237\; & 0.0370\; & 0.0553\; & 0.0818\; & 0.1238\; & 0.1912\; & 0.3023\; & 0.6190\; & 0 & 0\\ 0.0146\; & 0.0228\; & 0.0340\; & 0.0503\; & 0.0762\; & 0.1176\; & 0.1860\; & 0.3810\; & 1 & 1 \end{array}\right)$$


## 4. Conclusions

Submicron-sized quasi-spherical ZnO particles were prepared in a stirred media mill with chemical dissolution assistance from different solutions (i.e., HCl, CH_3_COOH, (COOH)_2_·2H_2_O, C_6_H_8_O_7_, NH_4_OH/NH_4_Cl and NaOH). The particles ground in a low-concentration (0.010 mol/L) CH_3_COOH solution under optimized parameters (i.e., grinding time of 30 min and solid content of 20 wt.%) had a decreased median size (i.e., d_50_ = 370 nm), a narrower size distribution (i.e., n = 2.28), enhanced sphericity (i.e., 0.91) and an increased specific surface area (i.e., 11.39 m^2^/g). This was mainly due to the effects of the chemical dissolution reaction and the shear/compressive stresses of beads on the particles’ surfaces in ultrafine grinding in stirred media mill for a short time. In addition, the chemical dissolution on the surfaces of particles ground in the CH_3_COOH solution occurred, thereby affecting the particle size, PSD and sphericity of the particles due to the mechanochemical effect. The grinding mechanism of ZnO particles analyzed by the selective and breakage functions revealed that the primary force for the particles’ breakage in the stirred media mill was shearing, and the secondary was compression. The addition of a CH_3_COOH solution could enhance the particle size and PSD. It is particularly effective in the reduction of the coarser particles.

## Figures and Tables

**Figure 1 materials-16-02558-f001:**
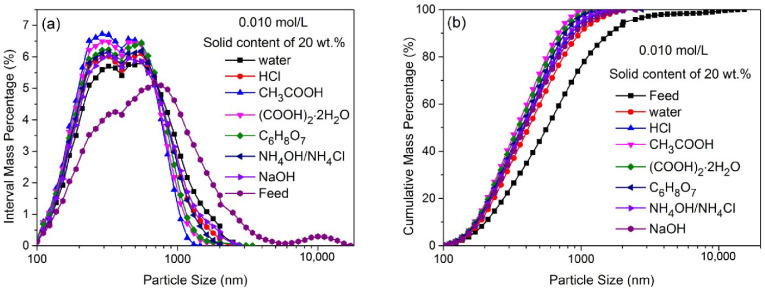
The PSD of ZnO particles ground in water or various other chemical solutions at a concentration of 0.010 mol/L for 30 min. (**a**) the interval mass distribution of ZnO particles; (**b**) the cumulative mass distribution of ZnO particles.

**Figure 2 materials-16-02558-f002:**
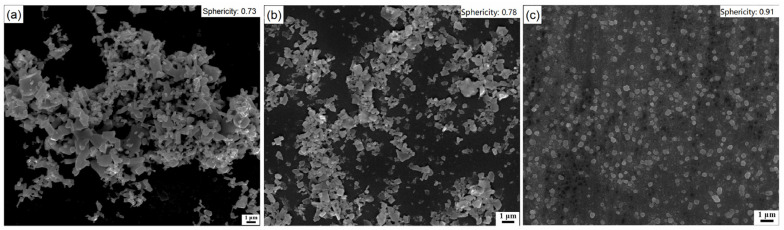
SEM pictures of zinc oxide particles: (**a**) feed; (**b**) ground for 30 min in water; (**c**) ground for 30 min in CH_3_COOH solution at a low concentration of 0.010 mol/L.

**Figure 3 materials-16-02558-f003:**
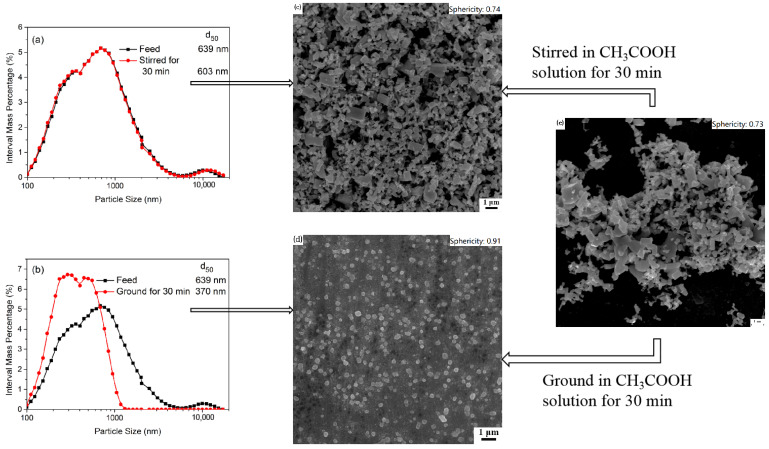
The PSD and SEM pictures of ZnO particles stirred or ground for 30 min in a CH_3_COOH solution. (**a**) the interval mass distribution of ZnO particles stirred for 30 min; (**b**) the interval mass distribution of ZnO particles ground for 30 min; (**c**) SEM image of ZnO particles stirred for 30 min; (**d**) SEM image of ZnO particlesground for 30 min; (**e**) SEM image of ZnO raw material.

**Figure 4 materials-16-02558-f004:**
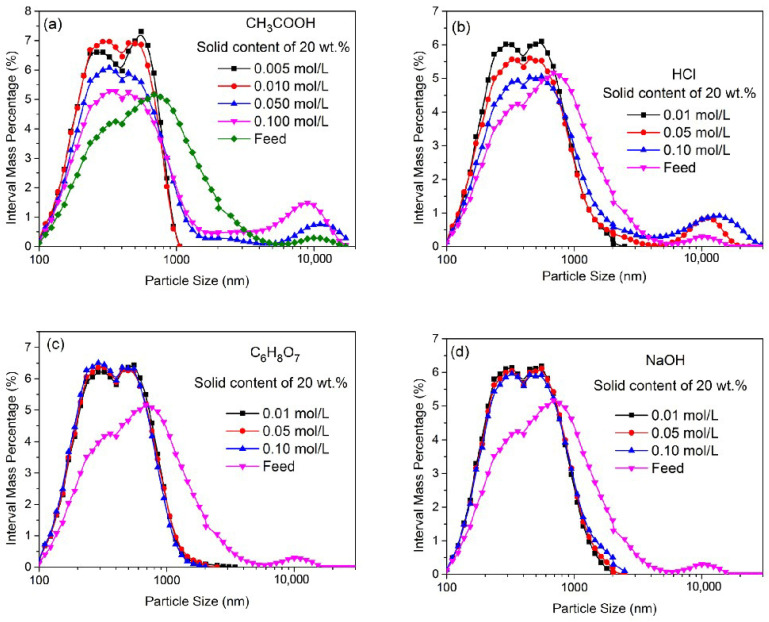
The PSD of ZnO particles ground for 30 min in CH_3_COOH, HCl, C_6_H_8_O_7_ and NaOH solutions at different concentrations. (**a**) the interval mass distribution of ZnO particles ground for 30 min in CH_3_COOH solutions at different concentrations; (**b**) the interval mass distribution of ZnO particles ground for 30 min in HCl solutions at different concentrations; (**c**) the interval mass distribution of ZnO particles ground for 30 min in C_6_H_8_O_7_ solutions at different concentrations; (**d**) the interval mass distribution of ZnO particles ground for 30 min in NaOH solutions at different concentrations.

**Figure 5 materials-16-02558-f005:**
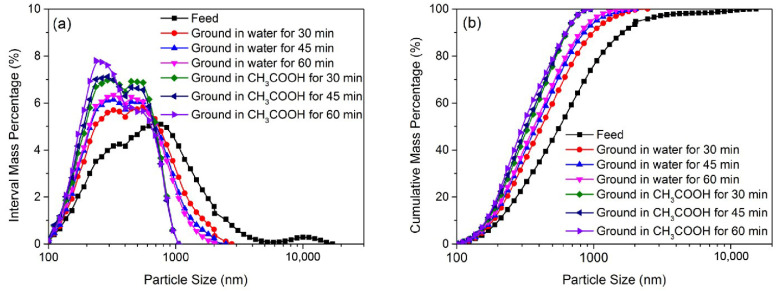
The PSD of ZnO particles ground in water or in a CH_3_COOH solution at a low concentration of 0.010 mol/L for different lengths of time. (**a**) the interval mass distribution of ZnO particles; (**b**) the cumulative mass distribution of ZnO particles.

**Figure 6 materials-16-02558-f006:**
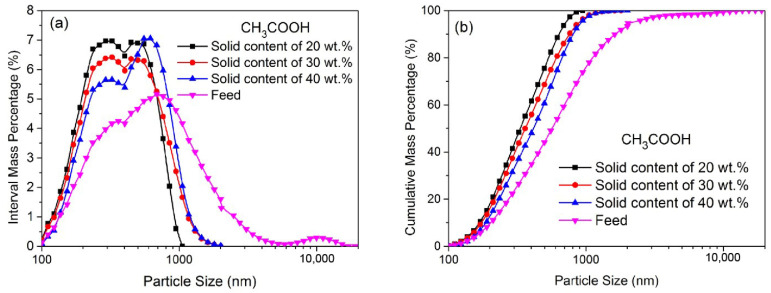
The PSD of ZnO particles ground for 30 min in CH_3_COOH solutions at a low concentration of 0.010 mol/L with different solid contents. (**a**) the interval mass distribution of ZnO particles; (**b**) the cumulative mass distribution of ZnO particles.

**Figure 7 materials-16-02558-f007:**
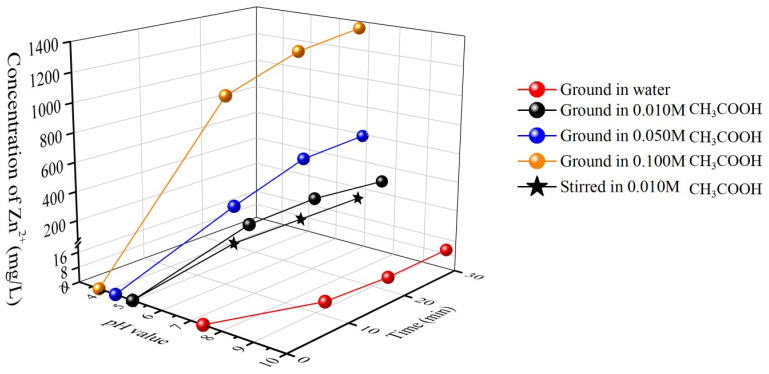
Concentrations of Zn^2+^ ions and pH values of zinc oxide particles ground for different lengths of time in CH_3_COOH solutions with different concentrations.

**Figure 8 materials-16-02558-f008:**
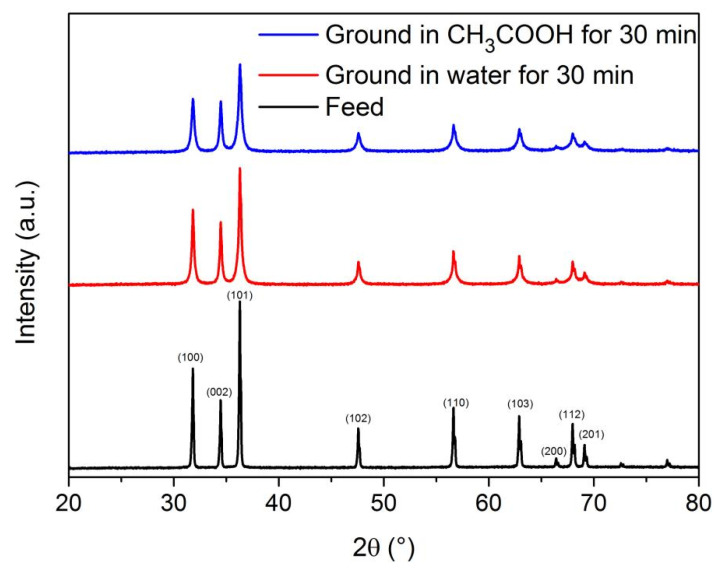
XRD patterns of feed and zinc oxide particles ground for 30 min in water or in a CH_3_COOH solution at a low concentration of 0.010 mol/L.

**Figure 9 materials-16-02558-f009:**
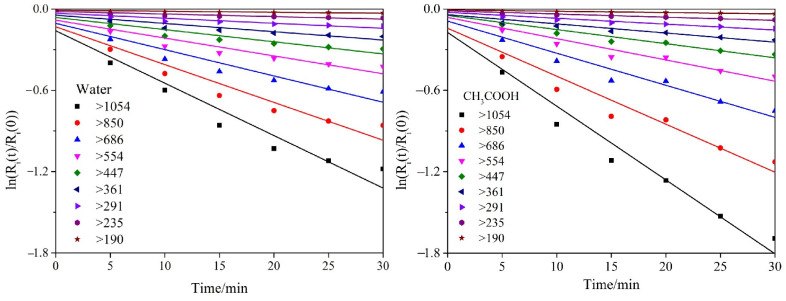
The first Kapur functions calculated by linear fitting of cumulative fraction residual of particles ground in water and a CH_3_COOH solution.

**Table 1 materials-16-02558-t001:** The different experimental factors and levels during the experiment.

Factors	Solution Type	Acid or Alkali Concentration (M or mol/L)	Solid Content (wt.%)	Grinding Time (min)
Levels	HCl CH_3_COOH (COOH)_2_·2H_2_O C_6_H_8_O_7_ NH_4_OH/NH_4_Cl NaOH Water	0.005 0.010 0.050 0.100	20 30 40	30 45 60

**Table 2 materials-16-02558-t002:** The d_50_, *n*, sphericity and specific surface area of ZnO particles ground for 30 min in water or one of various solutions at a concentration of 0.010 mol/L.

Solution Type	d_50_ (nm)	Uniformity Distribution Coefficient, *n*	Sphericity	Specific Surface Area (m^2^/g)
Feed	639	1.38	0.73	3.55
Water	453	1.64	0.78	9.77
HCl	426	1.92	0.83	9.87
CH_3_COOH	370	2.28	0.91	11.39
(COOH)_2_·2H_2_O	398	2.05	0.87	10.23
C_6_H_8_O_7_	409	1.94	0.85	10.10
NaOH	421	1.96	0.83	9.93
NH_4_OH/NH_4_Cl	433	1.85	0.83	9.85

**Table 3 materials-16-02558-t003:** The d_50_, *n* and sphericity values of ZnO particles ground for 30 min in the solutions of CH_3_COOH, C_6_H_8_O_7_, HCl and NaOH at different concentrations.

Solution Type	Concentration (mol/L)	d_50_ (nm)	Uniformity Distribution Coefficient, *n*	Sphericity
CH_3_COOH	0.005	382	2.14	0.89
CH_3_COOH	0.010	370	2.28	0.91
CH_3_COOH	0.050	424	0.84	0.86
CH_3_COOH	0.100	485	0.76	0.85
C_6_H_8_O_7_	0.010	409	1.94	0.85
C_6_H_8_O_7_	0.050	401	1.98	0.86
C_6_H_8_O_7_	0.100	390	2.01	0.87
HCl	0.010	426	1.92	0.83
HCl	0.050	459	0.83	0.81
HCl	0.100	537	0.75	0.80
NaOH	0.010	421	1. 96	0.83
NaOH	0.050	432	1.89	0.83
NaOH	0.100	439	1.83	0.84
Feed		639	1.38	0.73

**Table 4 materials-16-02558-t004:** The d_50_, *n* and sphericity values of ZnO particles ground in water or in a CH_3_COOH solution at a low concentration of 0.010 mol/L for different lengths of time.

Solution Type	Time (min)	d_50_ (nm)	Uniformity Distribution Coefficient, *n*	Sphericity
CH_3_COOH	30	370	2.28	0.91
CH_3_COOH	45	350	2.32	0.90
CH_3_COOH	60	336	2.38	0.87
Water	30	453	1.64	0.78
Water	45	425	1.93	0.80
Water	60	407	1.95	0.81

**Table 5 materials-16-02558-t005:** d_50_, *n* and sphericity values of ZnO particles ground for 30 min in CH_3_COOH solutions at a low concentration of 0.010 mol/L with different solid contents.

Solid Content (wt.%)	d_50_ (nm)	Uniformity Distribution Coefficient, *n*	Sphericity
20 wt.%	370	2.28	0.91
30 wt.%	402	1.90	0.85
40 wt.%	462	1.80	0.83

**Table 6 materials-16-02558-t006:** The ZnO particles size classes.

Size Class (i)	1	2	3	4	5	6	7	8	9	10
Size range (nm)	>1054	850–1054	686–850	554–686	447–554	361–447	291–361	235–291	190–235	<190

## Data Availability

Not applicable.

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
