# Peer review of "Chemical Dissolution-Assisted Ultrafine Grinding for Preparation of Quasi-Spherical Colloids of Zinc Oxide"

_materials, 2023, doi:10.3390/ma16072558_

Round 1

Reviewer 2 Report

The paper is focused on Chemical dissolution-assisted ultrafine grinding for preparation of quasi-spherical colloids of zinc oxide.

This work reports that submicron-size quase-spherical zinc oxide particles were prepared via high-energy density stirred media mill in diferent solutions, as water, hydrochloric acid, acetic acid, oxalic acid, citric acid, ammonium, mixed hydroxide/ammonium chloride and sodium hydroxide.

In according to the authors, there are several methods for the preparation of submicron-and- nano-sized spherical zinc oxide particles as precipitation, sol-gel, hydrothermal, emulsion,mechanochemical process and spray pyrolysis techinites. In addition, there are the physical methods as mechanical ultrafine grinding, plasma heating and laser ablation techniques. Among these methods, mechanical ultrafine grinding, used in this work, is a simply method and permits that irregular shape particles ground in stirred media mil become more regular with less sharp corners and edges. Thus, the authors should compare this method with the mentioned methods, in relation to the parameters obtained for the ZnO investigated in this work, such as medium size, distribution coefficient, sphericity and specific surface area of zinc oxide particles.

Authors must make the corrections:

Table 1 shows the experimental design for various parameters and the corresponding levels of each parameter. However, this Table must be completed with the parameters for the NH4OH/NH4Cl; NaOH and water.

The sphericity of ZnO particles was analyzed on the SEM images. However, the authors must also investigate the powder morphology of ZnO from TEM.

In results and discussion please revise the line 232:

“The particles ground in diferente solutions at the concentration of or 0.100 mol/L.”

Round 2

Reviewer 2 Report

The suggestions  and recommendations were considered by authors and the manuscript was modified accordingly.